# Deep Learning for Two-Sided Matching

## Abstract

We initiate the study of deep learning for the automated design of two-sided matching mechanisms. What is of most interest is to use machine learning to understand the possibility of new tradeoffs between *strategy-proofness* and *stability*. These properties cannot be achieved simultaneously, but the efficient frontier is not understood. We introduce novel differentiable surrogates for quantifying ordinal strategy-proofness and stability and use them to train differentiable matching mechanisms represented by neural networks that map discrete preferences to valid randomized matchings. We demonstrate that the efficient frontier characterized by these learned mechanisms is substantially better than that achievable through a convex combination of baselines of *deferred acceptance* (stable and strategy-proof for only one side of the market), *top trading cycles* (strategy-proof for one side, but not stable), and *randomized serial dictatorship* (strategy-proof for both sides, but not stable). This gives a new target for economic theory and opens up new possibilities for machine learning pipelines in matching market design.

## 1 Introduction

Two-sided matching markets, classically used for settings such as high-school matching, medical residents matching, and law clerk matching, and more recently used in online platforms such as Uber, Lyft, Airbnb, and dating apps, play a significant role in today's world. As a result, there is a significant interest in designing better mechanisms for two-sided matching.

The seminal work of Gale & Shapley (1962) introduces a simple mechanism for stable, one-to-one matching in two-sided markets—*deferred-acceptance* (DA)—which has been applied in many settings, including doctor-hospital matching (Roth & Peranson, 1999), school choice (Abdulkadiroğlu & Sönmez, 2003; Pathak & Sönmez, 2008; Abdulkadiroğlu et al., 2009), and cadet-matching (Sönmez & Switzer, 2013; Sönmez, 2013). The DA mechanism is *stable*, i.e., no pair of participants prefer each other to their match (or to being unmatched, if they are unmatched in the outcome). However, the DA mechanism is not *strategy-proof* (SP), and a participant can sometimes misreport their preferences to obtain a better outcome (although it is SP for participants on one side of the market). Although widely used, this failure of SP for the DA mechanism presents a challenge for two main reasons. First, it can lead to unfairness, where better-informed participants can gain an advantage in knowing which misreport strategies can be helpful. Second, strategic behavior can lead to lower quality, unintended outcomes, and outcomes that are unstable with respect to true preferences.

In general, it is well-known that there must necessarily be a tradeoff between stability and strategy-proofness: it is provably impossible for a mechanism to achieve both stability and strategy-proofness (Dubins & Freedman, 1981; Roth, 1982). A second example of a matching mechanism is *random serial dictatorship* (RSD) (Abdulkadiroglu & Sönmez, 1998), which is typically adopted for one-sided assignment problems rather than two-sided matching. When adapted to two-sided matching, RSD is SP but not stable. In fact, a participant may even prefer to remain unmatched than participate in the outcome of the matching. A third example of a matching mechanism is the *top trading cycles* (TTC) mechanism (Shapley & Scarf, 1974), also typically adopted for one-sided assignment problems rather than problems of two-sided matching. In application to two-sided matching, TTC is neither SP nor stable (although it is SP for participants on one side of the market).

There have been various research efforts to circumvent this impossibility result. Some relax the definition of strategyproofness (Mennle & Seuken, 2021) while others characterize the constraints under which stability and strategyproofness are achieved simultaneously (Kamada & Kojima, 2018;

Hatfield et al., 2021; Hatfield & Milgrom, 2005). The tradeoff between these desiderata remains poorly understood beyond the existing point solutions of DA, RSD, and TTC. However, we argue that real-world scenarios demand a more nuanced approach that considers both properties. The case of the Boston school choice mechanism highlights the negative consequences of lacking strategy-proofness, resulting in unfair manipulations by specific parents (Abdulkadiroglu et al., 2006). At the same time, the importance of stability in matching markets is well understood (Roth, 1991).

Recognizing this, and inspired by the success of deep learning in the study of revenue-optimal auction design (Duetting et al., 2019), we initiate the study of deep learning for the design of two-sided matching mechanisms. *We ask whether deep learning frameworks can enable a systematic study of this tradeoff.* By answering this question affirmatively, we open up the possibility of using machine learning pipelines to open up new opportunities for economic theory—seeking theoretical characterizations of mechanisms that can strike a new balance between strategyproofness and stability.

We use a neural network to represent the rules of a matching mechanism, mapping preference reports to a distribution over feasible matchings, and show how we can use an unsupervised learning pipeline to characterize the efficient frontier for the design tradeoff between stability and SP. The main methodological challenge in applying neural networks to two-sided matching comes from handling the ordinal preference inputs (the corresponding inputs are cardinal in auction design) and identifying suitable, differentiable surrogates for approximate strategy-proofness and approximate stability.

We work with randomized matching mechanisms, for which the strongest SP concept is *ordinal strategy-proofness*. This aligns incentives with truthful reporting, whatever an agent's utility function (i.e., for any cardinal preferences consistent with an agent's ordinal preferences). Ordinal SP is equivalent to the property of *first-order stochastic dominance* (FOSD) (Erdil, 2014), which suitably defines the property that an agent has a better chance of getting their top, top-two, top-three, and so forth choices when they report truthfully. As a surrogate for SP, we quantify during training the degree to which FOSD is violated. For this, we adopt an *adversarial learning approach*, augmenting the training data with defeating misreports that reveal the violation of FOSD. We also define a suitable surrogate to quantify the degree to which stability is violated. This surrogate aligns with the notion of *ex ante* stability—the strongest stability concept for randomized matching.

We propose two different neural network architectures to represent matching mechanisms — a simple fully connected neural network (MLP) and a convolutional neural network (CNN). Both architectures are trained with stochastic gradient descent (SGD) on loss functions that is based on various convex combinations of the two surrogate quantities. This allows us to construct the efficient frontier for stability and strategy-proofness for different market settings. Our main experimental results demonstrate that this novel use of deep learning can strike a much better trade-off between stability and SP than that achieved by a convex combination of the DA, TTC, and RSD mechanisms.

The CNN architecture, specifically designed for matching, uses $1 \times 1$ convolutions that treat inputs as two channels—each representing one side of the market's preferences. Since the number of sides is fixed, this architecture avoids the need to increase the number of hidden units (or filters, in the case of CNNs) as the input size grows. Moreover, the use of $1 \times 1$ convolutions ensures permutation equivariance, which narrows the search space, enhances generalization, and reduces training time by eliminating the need to calculate strategy-proofness violations for each agent individually. This architecture allows our approach to scale efficiently to markets with up to 50 agents on each side.

Taken as a whole, these results suggest that deep learning pipelines can be used to identify new opportunities for matching theory. For example, we identify mechanisms that are provably almost as stable as DA and yet considerably more strategy-proof. We also identify mechanisms that are provably almost as strategy-proof as RSD and yet considerably more stable. These discoveries raise opportunities for future work in economic theory, in regard to understanding the structure of these two-sided matching mechanisms as well as characterizing the preference distributions for which this is possible.

## 2 RELATED WORKS

Dubins & Freedman (1981) and Roth (1982) show the impossibility of achieving both stability and SP in two-sided matching. Alcalde & Barberà (1994) also show the impossibility of individually rational, Pareto efficient, and SP allocation rules, and this work has been extended to randomized

matching (Alva & Manjunath, 2020). RSD is SP but may not be stable or even individually rational (IR) (Abdulkadiroglu & Sönmez, 1998). We will see that the top trading cycles (TTC) mechanism (Shapley & Scarf, 1974), when applied in a two-sided context, is only SP for one side, and is neither stable nor IR. The DA mechanism (Gale & Shapley, 1962) is stable but not SP; see also (Roth et al., 1993), who study the polytope of stable matchings. The *stable improvement cycles* mechanism (Erdil & Ergin, 2008) achieves as much efficiency as possible on top of stability but fails to be SP even for one side of the market. Finally, a series of results show that DA becomes SP for both sides of the market in large-market limit contexts (Immorlica & Mahdian, 2015; Kojima & Pathak, 2009; Lee, 2016).

Aziz & Klaus (2019) discuss different stability and no envy concepts. We focus on *ex ante* stability (Kesten & Ünver, 2015), also discussed by (Roth et al., 1993) as *strong stability*. Mennle & Seuken (2021) discuss different notions of approximate strategy-proofness in the context of matching and allocation problems. In this work, we focus on ordinal SP and its analog of FOSD (Erdil, 2014). This is a strong and widely used SP concept in the presence of ordinal preferences. There are a lot of other desiderata, such as efficiency, that are also incompatible with strategyproofness. Mennle & Seuken (2017) study this trade-off through hybrid mechanisms which are convex combinations of a mechanism with good incentive properties with another which is efficient. In the context of social choice, other work studies the trade-off between approximate SP and desiderata, such as plurality and veto voting (Mennle & Seuken, 2016).

Conitzer & Sandholm (2002; 2004) introduced the automated mechanism design (AMD) approach that framed problems as a linear program. However, this approach faces severe scalability issues as the formulation scales exponentially in the number of agents and items (Guo & Conitzer, 2010). Overcoming this limitation, more recent work seeks to use deep neural networks to address problems of economic design (Duetting et al., 2019; Feng et al., 2018; Golowich et al., 2018; Curry et al., 2020; Shen et al., 2019; Rahme et al., 2020; Duan et al., 2022; Ivanov et al., 2022), but not until now to matching problems. As discussed in the introduction, two-sided matching brings about new challenges, most notably in regard to working with discrete, ordinal preferences and adopting the right surrogate loss functions for approximate SP and approximate stability. Other work has made use of support vector machines to search for stable mechanisms, but without considering strategy-proofness (Narasimhan et al., 2016). A different line of research is also considering stable matching together with bandits problems, where agent preferences are unknown *a priori* (Das & Kamenica, 2005; Liu et al., 2020; Dai & Jordan, 2021; Liu et al., 2022; Basu et al., 2021; Sankararaman et al., 2021; Jagadeesan et al., 2021; Cen & Shah, 2022; Min et al., 2022).

There have also been other recent efforts that leverage deep learning for matching (in the context of online bipartite matching (Alomrani et al., 2022)) and other related combinatorial optimization problems (Bengio et al., 2021). Most of these papers adopt a reinforcement learning based approach to compute their solutions. Our approach, on the other hand, is not sequential but rather end-to-end differentiable, and our parameter weights are updated through a single backward pass. Additionally, the focus of our work is on matching markets and mechanism design, and is concerned with capturing core economic concepts within a machine learning framework and balancing the trade-offs between stability and strategy-proofness.

## 3 PRELIMINARIES

Let $W$ denote a set of $n$ *workers* and $F$ denote a set of $m$ *firms*. A feasible *matching*, $\mu$, is a set of (worker, firm) pairs, with each worker and firm participating in at most one match. Let $\mathcal{B}$ denote the set of all *matchings*. If $(w, f) \in \mu$, then $\mu$ matches $w$ to $f$, and we write $\mu(w) = f$ and $\mu(f) = w$. If a worker or firm remains unmatched, we say it is matched to $\perp$. We also write $(w, \perp) \in \mu$ (resp. $(\perp, f) \in \mu$). Each worker has a *strict preference order*, $\succ_w$, over the set $\overline{F} = F \cup \{\perp\}$. Each firm has a strict preference order, $\succ_f$, over the set $\overline{W} = W \cup \{\perp\}$. Worker $w$ (firm $f$) prefers remaining unmatched to being matched with a firm (worker) ranked below $\perp$ (the agents ranked below $\perp$ are said to be *unacceptable*). If worker $w$ prefers firm $f$ to $f'$, then we write $f \succ_w f'$, similarly for a firm's preferences. Let $P$ denote the set of all *preference profiles*, with $\succ = (\succ_1, \ldots, \succ_n, \succ_{n+1}, \succ_{n+m}) \in P$ denoting a preference profile comprising of the preference order of the $n$ workers and then the $m$ firms.

A pair $(w, f)$ forms a *blocking pair for matching* $\mu$ if $w$ and $f$ prefer each other to their partners in $\mu$ (or $\perp$ in the case that one or both are unmatched). A matching $\mu$ is *stable* if and only if there are

no blocking pairs. A matching $\mu$ is *individually rational* (IR) if and only if it is not blocked by any individual; i.e., no agent finds its match unacceptable and prefers $\perp$.[1]

We work with *randomized matching mechanisms*, $g$, that map preference profiles, $\succ$, to distributions on matchings, denoted $g(\succ) \in \triangle(\mathcal{B})$ (the probability simplex on matchings). Let $r \in [0,1]^{(n+1) \times (m+1)}$ denote the *marginal probability*, $r_{wf} \geq 0$, with which worker $w$ is matched with firm $f$, for each $w \in \overline{W}$ and $f \in \overline{F}$. We require $\sum_{f' \in \overline{F}} r_{wf'} = 1$ for all $w \in W$, and $\sum_{w' \in \overline{W}} r_{w'f} = 1$ for all $f \in F$. For notational simplicity, we write $g_{wf}(\succ)$ for the marginal probability of matching worker $w$ (or $\perp$) and firm $f$ (or $\perp$).

**Theorem 1** (Birkhoff von-Neumann). *Given any randomized matching $r$, there exists a distribution on matchings, $\triangle(\mathcal{B})$, with marginal probabilities equal to $r$.*

The following definition is standard (Budish et al., 2013), and generalizes stability to randomized matchings.

**Definition 2** (Ex ante justified envy). A randomized matching $r$ causes *ex ante justified envy* if:

1. Some worker $w$ prefers $f$ over some fractionally matched firm $f'$ (including $f' = \perp$) and firm $f$ prefers $w$ over some fractionally matched worker $w'$ (including $w' = \perp$) ("$w$ has envy towards $w'$" and "$f$ has envy towards $f'$"), or
2. some worker $w$ finds a fractionally matched $f' \in F$ unacceptable, i.e. $r_{wf'} > 0$ and $\perp \succ_w f'$, or some firm $f$ finds a fractionally matched $w' \in W$ unacceptable, i.e. $r_{w'f} > 0$ and $\perp \succ_f w'$.

A randomized matching $r$ is *ex ante stable* if and only if it does not cause any *ex ante* justified envy. Ex ante stability reduces to the standard concept of stability for deterministic matching. Part (1) of the definition includes non-wastefulness: for any worker $w$, we should have $r_{w\perp} = 0$ if there exists some firm $f' \in F$ for which $r_{w'f'} > 0$, $w \succ_{f'} w'$ and $f' \succ_w \perp$ and for any firm $f$, we need $r_{\perp f} = 0$ if there exists some worker $w' \in W$ for which $r_{w'f'} > 0$, $f \succ_{w'} f'$ and $w' \succ_f \perp$. Part (2) of the definition captures IR: for any worker $w$, we should have $r_{wf'} = 0$ for all $f' \in F$ for which $\perp \succ_w f'$, and for any firm $f$, we need $r_{w'f} = 0$ for all $w' \in W$ for which $\perp \succ_f w'$.

To define strategy-proofness, say that $u_w : \overline{F} \to \mathbb{R}$ is a $\succ_w$-*utility* for worker $w$ when $u_w(f) > u_w(f')$ if and only if $f \succ_w f'$, for all $f, f' \in \overline{F}$. We similarly define a $\succ_f$-utility for a firm $f$. The following concept of ordinal SP is standard (Erdil, 2014), and generalizes SP to randomized matchings.

**Definition 3** (Ordinal strategy-proofness). A randomized matching mechanism $g$ satisfies *ordinal SP* if and only if, for all agents $i \in W \cup F$, for any preference profile $\succ$, and any $\succ_i$-utility for agent $i$, and for all reports $\succ'_i$, we have

$$\mathbf{E}_{\mu \sim g(\succ_i, \succ_{-i})}[u_i(\mu(i))] \geq \mathbf{E}_{\mu \sim g(\succ'_i, \succ_{-i})}[u_i(\mu(i))]. \tag{1}$$

By this definition, no worker or firm can improve their expected utility (for any utility function consistent with their preference order) by misreporting their preference order. For a deterministic mechanism, ordinal SP reduces to standard SP. Erdil (2014) shows that *first-order stochastic dominance* is equivalent to ordinal SP.

**Definition 4** (First Order Stochastic Dominance). A randomized matching mechanism $g$ satisfies *first order stochastic dominance* (FOSD) if and only if, for worker $w$, and each $f' \in \overline{F}$ such that $f' \succ_w \perp$, and all reports of others $\succ_{-w}$, we have (and similarly for the roles of workers and firms transposed),

$$\sum_{f \in F : f \succ_w f'} g_{wf}(\succ_w, \succ_{-w}) \geq \sum_{f \in F : f \succ_w f'} g_{wf}(\succ'_w, \succ_{-w}). \tag{2}$$

FOSD states that, whether looking at its most preferred firm, its two most preferred firms, or so forth, worker $w$ achieves a higher probability of matching on that set of firms for its true report than for any misreport. We make use of a quantification of the violation of this condition to provide a surrogate for the failure of SP during learning.

---

[1]Stability precludes empty matchings. For example, if a matching $\mu$ leaves a worker $w$ and a firm $f$ unmatched, where $w$ finds $f$ acceptable, and $f$ finds $w$ acceptable, then $(w, f)$ is a blocking pair to $\mu$.

**Theorem 5** ((Erdil, 2014))**.** *A two-sided matching mechanism is ordinal SP if and only if it satisfies FOSD.*

We consider three benchmark mechanisms: the stable but not SP *deferred-acceptance* (DA) mechanism, the SP but not stable *randomized serial dictatorship* (RSD) mechanism, and the *Top Trading Cycles* (TTC) mechanism, which is neither SP nor stable. The DA and TTC mechanisms are ordinal SP for the proposing side of the market but not for agents on both sides of the market. For a more detailed analysis of these mechanisms, refer to Appendix A.

## 4 TWO SIDED MATCHING AS A LEARNING PROBLEM

In this section, we develop the use of deep learning for the design of two-sided matching mechanisms.

### 4.1 NEURAL NETWORKS REPRESENTING MATCHING MECHANISMS

We use a neural network to represent a matching mechanism, designated as $g^\theta : P \to \triangle(\mathcal{B})$, parameterized by $\theta \in \mathbb{R}^d$. This network processes a preference profile as input and outputs a distribution over possible matchings. Below, we describe the method for representing these inputs and outputs, followed by a description of the network architecture.

**Inputs** To represent an agent's preference order in the input, we adopt a utility for each agent on the other side of the market that has a constant offset in utility across successive agents in the preference order. This is purely a representation choice and does not imply that we use this particular utility to study SP (on the contrary, we work with a FOSD-based quantification of the degree of approximation to ordinal SP). In particular, let $p_w^\succ = (p_{w1}^\succ, \dots, p_{wm}^\succ)$ and $q_f^\succ = (q_{1f}^\succ, \dots, q_{nf}^\succ)$ represent the preference order of a worker and firm, respectively. We define $p_{w\perp}^\succ = 0$ and $q_{\perp f}^\succ = 0$. Formally, we have

$$p_{wj}^\succ = \frac{1}{m}\Big(\mathbf{1}_{j\succ_w\perp} + \sum_{j'=1}^m (\mathbf{1}_{j\succ_w j'} - \mathbf{1}_{\perp\succ_w j'})\Big) \text{ and } q_{if}^\succ = \frac{1}{n}\Big(\mathbf{1}_{i\succ_f\perp} + \sum_{i'=1}^n (\mathbf{1}_{i\succ_f i'} - \mathbf{1}_{\perp\succ_f i})\Big)$$

where $\mathbf{1}_X$ is the indicator function for event $X$. To illustrate, we would represent this preference order $\succ$ with $w_1 : f_1, f_2, \perp, f_3$ as $p_{w_1}^\succ = (\frac{2}{3}, \frac{1}{3}, -\frac{1}{3})$. Taken together, the input is vector $(p_{11}^\succ, \dots, p_{nm}^\succ, q_{11}^\succ, \dots, q_{nm}^\succ)$ of $2 \times n \times m$ numbers.

**Outputs** The output of a valid matching network needs to be vector $r \in [0,1]^{n\times m}$, with $\sum_{j=1}^m r_{wj} \le 1$ and $\sum_{i=1}^n r_{if} \le 1$ for every every $w \in [n]$ and $f \in [m]$. This defines the marginal probabilities in a randomized matching for this input profile. To generate this output, the network first outputs two sets of scores $s \in \mathbb{R}_{\ge 0}^{(n+1)\times m}$ and $s' \in \mathbb{R}_{\ge 0}^{n\times(m+1)}$. These scores are constrained to be positive through the use of a softplus activation function in the last layer. We construct a boolean mask variable $\beta_{wf}$, which is 0 when the match is unacceptable to one or both the worker and firm, i.e., when $\perp \succ_w f$ or $\perp \succ_f w$, otherwise it is set to 1. We set $\beta_{n+1,f} = 1$ for $f \in F$ and $\beta_{w,m+1} = 1$ for $w \in W$. We multiply the scores $s$ and $s'$ element-wise with the corresponding mask variable to compute $\bar{s} \in \mathbb{R}_{\ge 0}^{(n+1)\times m}$ and $\bar{s}' \in \mathbb{R}_{\ge 0}^{n\times(m+1)}$. We normalize $\bar{s}$ along the rows and $\bar{s}'$ along the columns to obtain *normalized scores*, $\hat{s}$ and $\hat{s}'$ respectively. The match probability $r_{wf}$, for worker $w \in W$ and firm $f \in F$, is computed as the minimum of the normalized scores:

$$r_{wf} = \min\left(\frac{\bar{s}_{wf}}{\sum_{f'\in\overline{F}}\bar{s}_{wf'}}, \frac{\bar{s}'_{wf}}{\sum_{w'\in\overline{W}}\bar{s}'_{w'f}}\right).$$

Based on our construction, the allocation matrix $r$ is weakly doubly stochastic, with rows and columns summing to at most 1. Budish et al. (2013) show that any weakly doubly stochastic matrix can be decomposed to a convex combination of 0-1, weakly doubly stochastic matrices. Additionally, we have $r_{wf} = 0$ whenever $\beta_{wf} = 0$, ensuring that every matching in the support of the distribution will be IR.

**Model Architecture** We consider two different architectures in this paper. The first is the standard fully connected neural network (MLP) with $R$ fully connected hidden layers, each consisting of $J$ hidden units a *leaky ReLU* activation function. We use a fully connected output layer to produce the two set of scores $s, s'$ as described above.

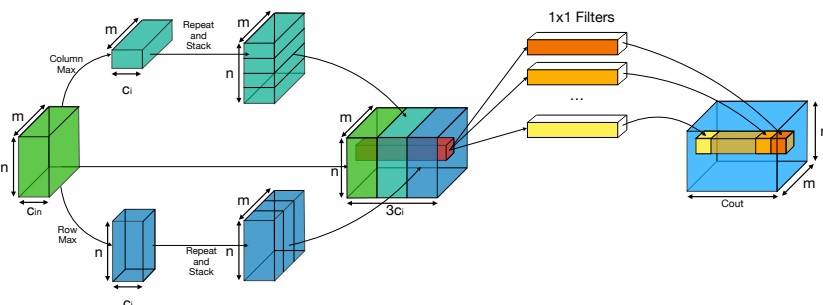

**Figure 1:** $1 \times 1$ Convolutions: For an input of size $n \times m$ with $c_{in}$ channels, row-wise and column-wise maximums are computed and stacked to the input as shown. Subsequently, $1 \times 1$ convolutions are applied using $c_{out}$ filters to produce the output.

We also introduce a Convolutional Neural Network (CNN) Architecture tailored for matching. The input to this CNN comprises of two channels, each comprising of an $n \times m$ matrix — $p_w^\succ$ for the worker preference and $q_w^\succ$ for the firm preference. Each layer within this architecture receives $c_{in}$ input channels (with $c_{in} = 2$ for the input layer) and produces 2 additional channels per each existing channel. The first of these additional channels captures the row-wise maximums at each index, while the second channel records the column-wise maximums. Following this, we apply $1 \times 1$ convolutions to preserve the original dimensions of $n \times m$ while expanding the depth of the output to $c_{out}$ channels, using $c_{out}$ filters. This method ensures the maintenance of spatial dimensions while enhancing feature representation through additional channels. See Figure 1 for more details.

We use $R$ convolutional layers with $J$ filters each. Note that we require this network to output two sets of scores $s, s'$ each having an additional column or row respectively (since $s \in \mathbb{R}^{(n+1) \times m}$ and $s' \in \mathbb{R}^{n \times (m+1)}$). To do this, we use 4 filters for the output layer. We compute the row-wise and column-wise mean of the penultimate output channel and append these to the first two layers as the additional row and column vector respectively. This operation yields the score sets $s$ and $s'$ effectively extending the dimensions to accommodate the required output format. See Figure 2 for more details.

A significant advantage of using CNNs is that the number of input channels to the network consistently remains at two, which means that the number of filters required does not vary significantly. Conversely, in a fully connected neural network architecture, the size of the input layer expands linearly with the number of workers and firms. Consequently, to achieve meaningful data representation, there is also a corresponding need to increase the number of hidden layers.

We also note that this implementation is an adaptation of the exchangeable matrix layer in Hartford et al. (2018) used to design permutation equivariant auctions (Rahme et al., 2020). Rather than employing the typical row-wise and column-wise *mean* calculations to compute the additional channels, our model utilizes the *max* operation. Since the *max* operation is still a commutative pooling operation, the permutation equivariance remains preserved (Ravanbakhsh et al., 2017).

Additionally, our benchmark mechanisms are permutation equivariant and we found that incorporating this property into the learned mechanisms offers several advantages. It enhances model complexity by reducing the search space and improves generalization as it inherently augments the training data by considering all permutations of the input data, effectively increasing the diversity of data the model is exposed to during training without actually expanding the minibatch size. Additionally, this symmetry ensures that the expected SP violation across all agents is the same, thereby reducing the need to individually compute the SP violations of all agents.

### 4.2 FORMULATION AS A LEARNING PROBLEM

We formulate a *loss function* $\mathcal{L}$ that is defined on training data of $\ell$ preference profiles, $D = \{\succ^{(1)}, \ldots, \succ^{(\ell)}\}$. Each preference profile $\succ$ sampled i.i.d. from a distribution on profiles. We allow for *correlated preferences*; i.e., workers may tend to agree that one of the firms is preferable to one of the other firms, and similarly for firms. The loss function captures a tradeoff between stability and ordinal SP. Recall that $g^\theta(\succ) \in [0,1]^{n \times m}$ denotes the randomized matching. We write $g_{w\perp}^\theta(\succ) = 1 - \sum_{f=1}^m g_{wf}^\theta(\succ)$ and $g_{\perp f}^\theta(\succ) = 1 - \sum_{w=1}^n g_{wf}^\theta(\succ)$ to denote the probability of worker $w$ and firm $f$ being unmatched, respectively.

**Figure 2:** CNN Architecture: Inputs $p^\succ, q^\succ$ are processed with $1 \times 1$ convolutional layers to produce scores $s, s'$. A Boolean mask $\beta$ is applied to these scores before normalization, ensuring that unacceptable matches in the final output $r$ have zero probability, thereby guaranteeing IR. Since all these operations are permutation equivariant, the final matching $r$ is also permutation equivariant.

**Stability Violation.** For worker $w$ and firm $f$, we define the *stability violation* at profile $\succ$ as

$$stv_{wf}(g^\theta, \succ) = \left( \sum_{w' \in \overline{W}} g^\theta_{w'f}(\succ) \cdot \max\{q^\succ_{wf} - q^\succ_{w'f}, 0\} \right) \cdot \left( \sum_{f' \in \overline{F}} g^\theta_{wf'}(\succ) \cdot \max\{p^\succ_{wf} - p^\succ_{wf'}, 0\} \right)$$

This captures the first kind of *ex ante* justified envy in Definition 2. We can omit the second kind of *ex ante* justified envy because the learned mechanisms satisfy IR through the use of masked softmax (and thus, there are no violations of the second kind).

The average stability violation (or just *stability violation*) of mechanism $g^\theta$ on profile $\succ$ is $stv(g^\theta, \succ) = \frac{1}{2} \left( \frac{1}{m} + \frac{1}{n} \right) \sum_{w=1}^n \sum_{f=1}^m stv_{wf}(g^\theta, \succ)$. We define the *expected stability violation*, $STV(g^\theta) = \mathbb{E}_\succ stv(g^\theta, \succ)$. We also write $stv(g^\theta)$ to denote the average stability violation on the training data.

**Theorem 6.** *A randomized matching mechanism $g^\theta$ is ex ante stable up to zero-measure events if and only if $STV(g^\theta) = 0$.*

**Ordinal SP violation.** We turn now to quantifying the degree of approximation to ordinal SP. Let $\succ_{-i} = (\succ_1, \ldots, \succ_{i-1}, \succ_{i+1}, \ldots, \succ_{n+m})$. For a valuation profile, $\succ \in P$, and a mechanism $g^\theta$, let $\Delta_{wf}(g^\theta, \succ'_w, \succ) = g^\theta_{wf}(\succ'_w, \succ_{-w}) - g^\theta_{wf}(\succ_w, \succ_{-w})$. The *regret* to worker $w$ is defined as:

$$\text{regret}_w(g^\theta, \succ) = \max_{\succ'_w \in P} \left( \max_{f' \succ_w \perp} \sum_{f \succ_w f'} \Delta_{wf}(g^\theta, \succ'_w, \succ) \right) \tag{3}$$

**Theorem 7.** *The regret to a worker (firm) for a given preference profile is the maximum amount by which the worker (firm) can increase their expected normalized utility through a misreport, fixing the reports of others.*

The *expected regret* for a mechanism, $RGT(g^\theta)$, is simply the expected regret over all agents over all profiles. We can also write $rgt(g^\theta)$ to denote the average regret on training data.

**Theorem 8.** *A randomized mechanism, $g^\theta$, is ordinal SP up to zero-measure events if and only if $RGT(g^\theta) = 0$.*

**Training Procedure.** For a mechanism parameterized as $g^\theta$, the training problem that we formulate is,

$$\min_\theta \ \lambda \cdot stv(g^\theta) + (1 - \lambda) \cdot rgt(g^\theta) \tag{4}$$

where $\lambda \in [0, 1]$ controls the tradeoff between approximate stability and approximate SP. We use SGD to minimize Equation (4), utilizing fresh minibatch of preferences sampled online for each update. The gradient of the degree of violation of stability with respect to network parameters is straightforward to calculate. The gradient of regret is complicated by the nested maximization in the definition of regret. In order to compute the gradient, we first solve the inner maximization by checking possible misreports. Let $\hat{\succ}_i^{(\ell)}$ denote the *defeating preference report* for agent $i$ (a worker or firm) at preference profile $\succ^{(\ell)}$ that maximizes $regret_i(g^\theta, \succ^{(\ell)})$. Given this, we obtain the derivative of regret for agent $i$ with respect to the network parameters, fixing the misreport to the defeating valuation and adopting truthful reports for the others.

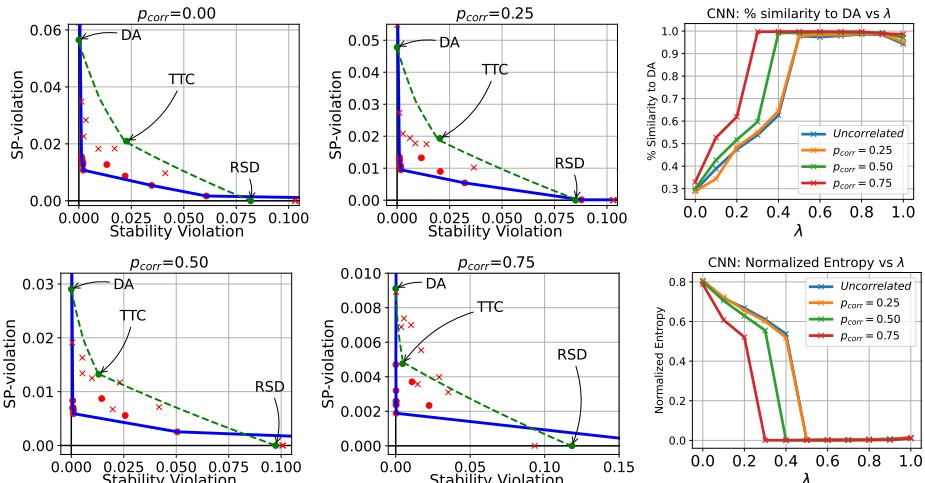

**Figure 3:** Comparing stability violation and strategy-proofness violation from the learned mechanisms for different choices of $\lambda$ (red dots and crosses are points learned by the CNN and MLP architecture respectively) with the best of worker- and firm-proposing DA, as well as TTC, and RSD, in $4 \times 4$ two-sided matching, and considering uncorrelated preference orders (Setting A) well as markets with increasing correlation (Setting B) with $p_{\text{corr}} \in \{0.25, 0.5, 0.75\}$). The stability violation for TTC and RSD includes IR violations. **Top, Right**: Comparing the average instance-wise max similarity scores ($sim(g^{\theta})$) of the learned mechanisms (using the CNN architecture) with worker- and firm-proposing DA. **Bottom, Right**: Normalized entropy of the learned mechanisms (using CNN architecture) for different values of the tradeoff parameter $\lambda$.

## 5 EXPERIMENTAL RESULTS

We study the following market settings:

- A. For *uncorrelated preferences*, for each worker or firm, we sample uniformly at random from all preference orders, and then, with probability, $p_{\text{trunc}} = 0.2$ (truncation probability), we choose at random a position at which to truncate this agent's preference order.
- B. For *correlated preferences*, we sample a preference profile as in the uncorrelated case. We also sample a common preference order on firms and a common preference order on workers. For each agent, with probability, $p_{\text{corr}} > 0$, we replace its preference order with the common preference order for its side of the market.

Specifically, we consider matching problems with $n = 4$ workers and $m = 4$ firms with uncorrelated preference and varying probability of correlation $p_{\text{corr}} = \{0.25, 0.5, 0.75\}$.

We report the results on a test set of 204,800 preference profiles, and use the AdamW optimizer to train our models. We use the *PyTorch* deep learning library, and all experiments are run on a single A100 or H100 NVIDIA GPU. Please refer to Appendix E for additional details.

We compare the performance of our mechanisms, varying parameter $\lambda$ between $0$ and $1$, with the best of worker- and firm- proposing DA and TTC (as determined by average SP violation over the test data) and RSD[2]. We also compare against convex combinations of DA, TTC, and RSD. We plot the resulting frontier on stability violation ($stv(g^{\theta})$) and SP violation ($rgt(g^{\theta})$) in Figure 3. TTC and RSD mechanisms do not guarantee IR, so we include the IR violations in the reported stability violation (none of the other mechanisms fail IR). We define the IR violation at profile $\succ$ as:

$$irv(g, \succ) = \frac{1}{2m} \sum_{w=1}^{n} \sum_{f=1}^{m} g_{wf}(\succ) \cdot (\max\{-q_{wf}, 0\}) + \frac{1}{2n} \sum_{w=1}^{n} \sum_{f=1}^{m} g_{wf}(\succ) \cdot (\max\{-p_{wf}, 0\})$$

At $\lambda = 0.0$, we learn a mechanism that has very low regret ($\approx 0$) but poor stability. This performance is similar to that of RSD. For large values of $\lambda$, we learn a mechanism that approximates DA. For intermediate values, we find solutions that dominate the convex combination of DA, TTC, and RSD and find novel and interesting tradeoffs between SP and stability. Notably, for lower levels of

---

[2]We only plot the performance of one-sided RSD as it achieves lower stability violation the two-sided version

correlations we see substantially better SP than DA along with very little loss in stability. Given the importance of stability in practice, this is a very intriguing discovery. For higher levels of correlations, we see substantially better stability than RSD along with very little loss in SP. It is also interesting to see that TTC itself has intermediate properties, between those of DA and RSD. Comparing the scale of the y-axes, we can also see that increasing correlation tends to reduce the opportunity for strategic behavior across both the DA and the learned mechanisms.

In interpreting the rules of the learned mechanisms, and considering the importance of DA, we can also compare their functional similarity with DA. For this, let $w$-DA and $f$-DA denote the worker- and firm-proposing DA, respectively. For a given preference profile, we compute the similarity of the learned rule with DA as

$$
sim(g^\theta, \succ) = \max_{\mathcal{M} \in \{w\text{-DA}, f\text{-DA}\}} \frac{\sum_{(w,f): g_{wf}^{\mathcal{M}}(\succ)=1} g_{wf}^\theta(\succ)}{\sum_{(w,f): g_{wf}^{\mathcal{M}}(\succ)=1} 1}. \tag{5}
$$

This calculates the agreement between the two mechanisms, normalized by the size of the DA matching, and taking the best of $w$-DA or $f$-DA. Let $sim(g^\theta)$ denote the average similarity score on test data. As we increase $\lambda$, i.e., penalize stability violations more, we see in Figure 3 (Top, Right) that the learned matchings get increasingly close to the DA matchings, as we might expect. We also quantify the degree of randomness of the learned mechanisms, by computing the *normalized entropy per agent*, taking the expectation over all preference profiles. For a given profile $\succ$, we compute normalized entropy per agent as (this is 0 for a deterministic mechanism):

$$
H(\succ) = -\frac{1}{2n} \sum_{w \in W} \sum_{f \in \overline{F}} \frac{g_{wf}(\succ) \log_2 g_{wf}(\succ)}{\log_2 m} - \frac{1}{2m} \sum_{f \in F} \sum_{w \in \overline{W}} \frac{g_{wf}(\succ) \log_2 g_{wf}(\succ)}{\log_2 n}. \tag{6}
$$

Figure 3 (Bottom, Right) shows how the entropy changes with $\lambda$. As we increase $\lambda$ and the mechanisms come closer to DA, the allocations of the learned mechanisms also becomes less stochastic. In Appendix F, we present additional experiments to show the the expected welfare vary for the different learned mechanisms.

**Scaling** Note that since ordinal preferences are discrete, the computation of SP violations involves enumeration of all possible misreports. We resolve the challenge that this presents in scaling to a larger numbers of agents by assuming a suitable structure on preference orderings and misreports in the domain. Indeed, in situations where there is uncertainty regarding preferences, small support is commonly expected and observed in real world data (Drummond & Boutilier, 2014; Hazon et al., 2012). With this consideration, we have designed the following market setting:

C. Each worker or firm's preferences are sampled uniformly at random from a dataset of $\ell$ preference orders, with a truncation probability, $p_{\text{trunc}} = 0.2$. We assume the dataset is *public*. The agents can choose to misreport by selecting any preference from the dataset or by truncating them.

D. Each worker or firm's preferences are sampled uniformly at random from a dataset of $\ell$ preference orders, with a truncation probability, $p_{\text{trunc}} = 0.20$. We assume the dataset is *private* and the agents can only choose to misreport by truncating their own preferences.

For the public setting C, we set $n = m = 10$ and $\ell = 10$, allowing for up to 100 possible preference orders per agent when including truncations. The number of possible misreports per agent is 100 as well. For the private setting D, we consider two cases with $n = m = 20$ and $n = m = 50$, each with $\ell = 10$, allowing for up to 200 and 500 possible preference orders respectively per agent. The number of possible misreports in the private setting is restricted to 20 and 50 respectively. For these settings, we only use the CNN architecture.

It is important to note that, in these settings, computing the ex ante representation of Randomized Serial Dictatorship (RSD) — which involves $(n + m)!$ priority orders — proves to be intractable. This is required for computing the stability violation. Therefore, we use Serial Dictatorship (SD) with a fixed priority order as our baseline.

The results of these experiments are presented in Figure 4. Our findings indicate that the learned mechanisms outperform the combined benchmarks of deferred acceptance (DA) and randomized serial dictatorship (SD) across various configurations of the tradeoff parameter $\lambda$. For larger values

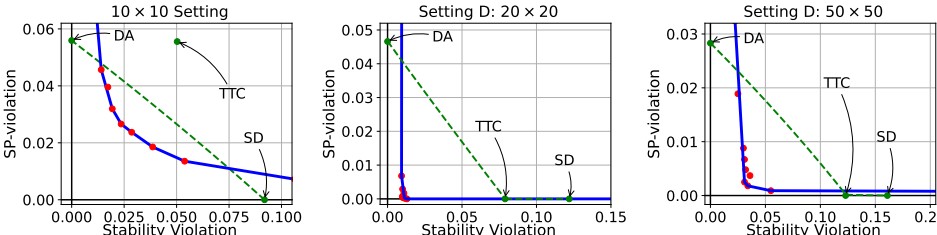

**Figure 4:** Comparing the Stability and SP violations for different choices of $\lambda$ for Setting C (public) with $n = m = 10$ (left) and Setting D (private) with $n = m = 20$ (middle) and $n = m = 50$ (right) using the CNN architecture.

of $\lambda$, our mechanisms do not closely approximate DA (unlike the results for our previous setting). This suggests the existence of multiple stable mechanisms when we consider a restricted preference domain. Additionally, note that we use the linear scalarization (Equqation [4]) to compute different points on the frontier. While this method is commonly used because of its simplicity, it occasionally struggles with convergence to a diverse set of solutions Lin et al. (2019). A finer adjustment of $\lambda$ within the range $[0.9, 1.0)$ could potentially reveal mechanisms more closely aligned with DA.

## 6    DISCUSSION

The methodology and results in this paper give a first but crucial step towards using machine learning to understanding the structure of mechanisms that achieve nearly the same stability as DA while surpassing DA in terms of strategy-proofness. This is an interesting observation, given the practical and theoretical importance of the DA mechanism. There are other interesting questions waiting to be addressed. For instance, can we use this kind of framework to understand other tradeoffs, such as tradeoffs between strategy-proofness and efficiency?

As discussed previously, a challenge in scaling to larger problems is the need to find defeating misreports, as exhaustively enumerating all misreports for an agent becomes intractable as the number of agents on the other side of the market increases. A simple remedy that we adopted here is to work in domains where there exists some structure on the preference domain, so that not all possible preference orders exist; e.g., *single-peaked preferences* are an especially stark example (Black, 1948). Another remedy is to restrict the language available to agents in making preference reports; e.g., it is commonplace to only allow for "top-$k$ preferences" to be reported. It will also be interesting to study complementary approaches that relax the discrete set of preference orderings to a continuous convex hull such as the *Birkhoff polytope* and using gradient ascent to identify misreports. Despite this limitation, our current approach scales much further than other, existing methods for automated design, which are not well suited for this problem. For instance, methods that use linear programs or integer programs do not scale well because of the number of variables required to make explicit the input and output structure of the functional that must be optimized over.

A second challenge is that we have not been able to find a suitable, publicly available dataset to test our approach. As a fallback, we have endeavored to capture some real-world structures by varying the correlation between agent preferences and the truncation probabilities of preferences. Using such stylized, probabilistic models and simulations for validating approaches is a well-established and prevalent practice, consistently utilized when investigating two-sided matching markets (Chen & Sönmez, 2006; Echenique & Yariv, 2013; Das & Kamenica, 2005; Liu et al., 2020; Dai & Jordan, 2021). For instance, Chen and Sönmez (Chen & Sönmez, 2006) design an environment for school choice where they consider six different schools with six seats each and where the students' preferences are simulated to depend on proximity, quality, and a random factor. Echenique an Yariv (Echenique & Yariv, 2013) use a simulation study with eight participants on each side of the market, with the payoff matrix designed such that there are one, two, or three stable matches. Further, recent papers on bandit models for stable matching model agent preferences through synthetic datasets (Das & Kamenica, 2005; Liu et al., 2020; Dai & Jordan, 2021).

In closing, we see exciting work ahead in advancing the design of matching mechanisms that strike the right balance between stability, strategyproofness, and other considerations that are critical to real-world applications. As an example, it will be interesting to extend the learning framework to encompass desiderata such as capacity limitations or fairness considerations.

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

## A   DEFERRED ACCEPTANCE, RSD, AND TTC.

We consider three benchmark mechanisms: the stable but not SP *deferred-acceptance* (DA) mechanism, the SP but not stable *randomized serial dictatorship* (RSD) mechanism, and the *Top Trading Cycles* (TTC) mechanism, which is neither SP nor stable. The DA and TTC mechanisms are ordinal SP for the proposing side of the market but not for agents on both sides of the market.

**Definition 9** (Deferred-acceptance (DA))**.** In *worker-proposing deferred-acceptance* (firm-proposing is defined analogously), each worker $w$ maintains a list of acceptable firms ($f \succ_w \perp$) for which it has not had a proposal rejected ("remaining firms"). Repeat until all proposals are accepted:

- $\forall w \in W$: $w$ proposes to its best acceptable, remaining firm.
- $\forall f \in F$: $f$ tentatively accepts its best proposal (if any), and rejects the rest.
- $\forall w \in W$: If $w$ is rejected by firm $f$, it updates its list of acceptable firms to remove $f$.

**Theorem 10** (see (Roth & Sotomayor, 1990)). *DA is stable but not Ordinal SP.*

**Definition 11** (Randomized serial dictatorship (RSD)). In the two-sided version of RSD, we first sample a *priority order*, $\pi$, on the set $W \cup F$, uniformly at random, such that $\pi = (\pi_1, \pi_2, \ldots, \pi_{m+n})$ is a permutation on $W \cup F$ in decreasing order of priority. For the one-sided version, we sample a priority order $\pi$ on either $W$ or $F$.

Proceed as follows:

- Initialize matching $\mu$ to the empty matching.
- In round $k = 1, \ldots, |\pi|$:
  - If $\pi_k$ is not yet matched in $\mu$, then add to $\mu$ the match between $\pi_k$ and its most preferred unmatched agent, or $\perp$ if all remaining agents are unacceptable to $\pi_k$.

**Theorem 12.** *RSD satisfies FOSD—and thus is ordinal SP by Theorem 5—but is not stable.*

*Proof.* We first show RSD satisfies FOSD and is thus ordinal SP. Consider agent $i$ in some position $k$ in the order. The agent's report has no effect on the choices of preceding agents, whether workers or firms (including whether agent $i$ is selected by an agent on the other side). Reporting its true preference ensures, in the event that it remains unmatched by position $k$, that it is matched with its most preferred agent of those remaining. For the one-sided version, the same argument holds for agents that are in the priority order. If an agent isn't on the side that's on the priority order, then that agent's report has no effect at all.

In the following example, we show RSD mechanism is not stable.

**Example 13.** Consider $n = 3$ workers and $m = 3$ firms with the following preference orders:

$$w_1 : f_2, f_3, f_1, \perp \quad f_1 : w_1, w_2, w_3, \perp$$
$$w_2 : f_2, f_1, f_3, \perp \quad f_2 : w_2, w_3, w_1, \perp$$
$$w_3 : f_1, f_3, f_2, \perp \quad f_3 : w_3, w_1, w_2, \perp$$

The matching found by worker-proposing DA is $(w_1, f_3), (w_2, f_2), (w_3, f_1)$. This is a stable matching. If $f_1$ truncates and misreports its preference as $f_1 : w_1, w_2, \perp, w_3$, the matching found is $(w_1, f_1), (w_2, f_2), (w_3, f_3)$. Firm $f_1$ is matched with a more preferred worker, and hence the mechanism is not strategy-proof. Now consider the matching under RSD. The marginal matching probabilities $r$ is given by:

$$r = \begin{pmatrix} \frac{11}{24} & \frac{1}{4} & \frac{7}{24} \\ \frac{1}{6} & \frac{3}{4} & \frac{1}{12} \\ \frac{3}{8} & 0 & \frac{5}{8} \end{pmatrix}$$

$f_2$ and $w_2$ are the most preferred options for $w_2$ and $f_2$ respectively and they would prefer to be matched with each other always rather than being fractionally matched with each other. Here $(w_2, f_2)$ is a blocking pair and thus RSD is not stable.

$\square$

**Definition 14** (Top Trading Cycles (TTC)). In *worker-proposing TTC* (firm-proposing is defined analogously), each agent (worker or firm) maintains a list of acceptable firms. Repeat until all agents are matched:

- Form a directed graph with each unmatched agent pointing to their most preferred option. The agents can point at themselves if there are no acceptable options available. Every worker that is a part of a cycle is matched to a firm it points to ( or itself, if the worker is pointing at itself). The unmatched agents remove from their lists every matched agent from this round.

**Theorem 15.** *TTC is neither strategy-proof nor stable for both sides.*

*Proof.* The following example shows that TTC is neither strategyproof nor stable.

**Example 16.** Consider $n = 4$ workers and $m = 4$ firms with the following preference orders:

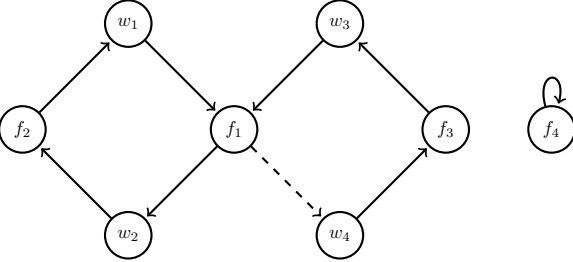

**Figure 5:** Round 1 of TTC. The solid lines represent workers and firms pointing to their top preferred agent truthfully. The dashed line represents a misreport by $f_1$

$$
\begin{aligned}
&w_1 : f_1, \perp && f_1 : w_2, w_3, w_4, \perp \\
&w_2 : f_2, \perp && f_2 : w_1 \perp \\
&w_3 : f_1, \perp && f_3 : w_3 \perp \\
&w_4 : f_3, \perp && f_4 : \perp
\end{aligned}
$$

If all agents report truthfully, $w_1$ is matched with $f_1$. This violates IR as $\perp \succ_{f_1} w_1$ and thus the matching is not ex-ante stable. If $f_1$ misreports its preference as $f_1 : w_4, w_3, \perp$, then $w_3$ is matched with $f_1$. Since $f_1$ is matched with a more preferred worker $w_3$ with $w_3 \succ_{f_1} \perp \succ_{f_1} w_1$, TTC is not strategyproof.

$\square$

**Remark 17.** TTC, like RSD, is usually used in one-sided assignment problems, where it is SP, and where the notion of stability which is an important consideration in two-sided matching, is not a concern.

## B    PROOF OF THEOREM 6

**Theorem 6.** *A randomized matching mechanism $g^\theta$ is ex ante stable up to zero-measure events if and only if $STV(g^\theta) = 0$.*

*Proof.* Since $stv(g^\theta, \succ) \geq 0$ then $STV(g^\theta) = \mathbb{E}_\succ stv(g^\theta, \succ) = 0$ if and only if $stv(g^\theta, \succ) = 0$ except on zero measure events. Moreover, $stv(g^\theta, \succ) = 0$ implies $stv_{wf}(g^\theta, \succ) = 0$ for all $w \in W$, all $f \in F$. This is equivalent to no justified envy. For firm $f$, this means $\forall w' \neq w$, $q_{wf}^\succ \leq q_{w'f}^\succ$ if $g_{w'f}^\theta > 0$ and $q \succ_{wf} \leq 0$ if $g_{\perp f}^\theta > 0$. Then there is no justified envy for a firm $f$. Analogously, there is no justified envy for worker $w$. If $g^\theta$ is *ex ante* stable, it trivially implies $STV(g^\theta) = 0$ by definition. $\square$

## C    PROOF OF THEOREM 7

**Theorem 7.** *The regret to a worker (firm) for a given preference profile is the maximum amount by which the worker (firm) can increase their expected normalized utility through a misreport, fixing the reports of others.*

*Proof.* Consider some worker $w \in W$. Without loss of generality, let $\succ_w: f_1, \ldots, f_k, \perp, f_{k+1}, \ldots, f_m$. Any normalized $\succ_w$-*utility* function, $u_w$, consistent with ordering given by $\succ_w$ satisfies $1 \geq u_w(f_1) \geq u_w(f_2) \geq \ldots u_w(f_k) \geq 0 \geq u_w(f_{k+1}) \geq \ldots u_w(f_m)$. Let $U_w$ be the set of all such consistent utility functions.

Consider some misreport $\succ_w'$. We have $\Delta_{wf}(g^\theta, \succ_w', \succ) = g_{wf}^\theta(\succ_w', \succ_{-w}) - g_{wf}^\theta(\succ_w, \succ_{-w})$. The increase in utility for worker $w$ when the utility function is $u_w$ is given by $\sum_{f \in F} u_w(f) \Delta_{wf}(g^\theta, \succ_w', \succ)$. The maximum amount by which worker $w$ can increase their expected normalized utility through misreport $\succ_w'$ is given by the objective: $\max_{u_w \in U_w} \sum_{f \in F} u_w(f) \Delta_{wf}(g^\theta, \succ_w', \succ)$.

Since $g^\theta$ always guarantees IR, we have:

$$\sum_{f \in F: \perp \succ f} u_w(f) \Delta_{wf}(g^\theta, \succ_w', \succ) = \sum_{f \in F: \perp \succ_w f} u_w(f) g_{wf}^\theta(\succ_w', \succ_{-w}) \leq 0 \tag{7}$$

. Thus, we can simplify our search space by only considering $u_w \in U_w$ where $u_w(f_{k+1}), \ldots, u_w(f_m) = 0$.

Define $\delta_k = u_w(f_k), \delta_{k-1} = u_w(f_{k-1}) - u_w(f_k), \delta_1 = u_w(f_1) - u_w(f_2)$. This objective can thus be rewritten as:

$$\max \sum_{f=1}^{k} \left( \sum_{i=f}^{k} \delta_i \right) \Delta_{wf}(g^\theta, \succ_w', \succ) \tag{8}$$

$$\text{such that } \sum_{i=1}^{k} \delta_i \leq 1 \text{ and } \delta_1, \ldots \delta_k \geq 0 \tag{9}$$

Changing the order of summation, we have the following optimization problem:

$$\max \sum_{i=1}^{k} \delta_i \left( \sum_{f=1}^{i} \Delta_{wf}(g^\theta, \succ_w', \succ) \right) \tag{10}$$

$$\text{such that } \sum_{i=1}^{k} \delta_i \leq 1 \text{ and } \delta_1, \ldots \delta_k \geq 0 \tag{11}$$

This objective is of the form $\max_{\|x\|_1 \leq 1} x^T y$ and it's solution is given by the $\|y\|_\infty$. Thus, the solution to the above maximization problem is given by $\max_{i \in [k]} \sum_{f=1}^{i} \Delta_{wf}(g^\theta, \succ_w', \succ)$. But this is the same as $\max_{f': f' \succ_w \perp} \sum_{f: f \succ_w f'} \Delta_{wf}(g^\theta, \succ_w', \succ)$. Computing the maximum possible increase over all such misreports gives us $\max_{\succ_w' \in P} \left( \max_{f': f' \succ_w \perp} \sum_{f: f \succ_w f'} \Delta_{wf}(g^\theta, \succ_w', \succ) \right)$. This quantity is exactly $regret_w(g^\theta, \succ)$. The proof follows similarly for any firm $f$. $\qquad \square$

# D    PROOF OF THEOREM 8

**Theorem 8.** *A randomized mechanism, $g^\theta$, is ordinal SP up to zero-measure events if and only if $RGT(g^\theta) = 0$.*

*Proof.* Since $regret(g^\theta, \succ) \geq 0$ then $RGT(g^\theta) = \mathbb{E}_\succ regret(g^\theta, \succ) = 0$ if and only if $regret(g^\theta, \succ) = 0$ except on zero measure events. Moreover, $regret(g^\theta, \succ) = 0$ implies $regret_w(g^\theta, \succ) = 0$ for any worker $w$ and $regret_f(g^\theta, \succ) = 0$ for any firm $f$. Thus, the maximum utility increase on misreporting is at most zero, and hence $g^\theta$ is ordinal SP. If $g^\theta$ is Ordinal-SP, it is also satisfies FOSD 5 and it is straightforward to show that $regret(g^\theta, \succ) = 0$. $\qquad \square$

# E    TRAINING DETAILS AND HYPERPARAMETERS

For the MLP architecture, we use $R = 4$ hidden layers with 256 hidden units each for all settings A and B. We use the leaky ReLU activation function at each of these layers. To train our neural network, we use the AdamW Optimizer with decoupled weight delay regularization (implemented as *AdamW* optimizer in PyTorch) We set the learning rate to $0.005$ for uncorrelated preferences setting and $0.002$ when $p_{corr} = \{0.25, 0.5, 0.75\}$. The remaining hyperparameters of the optimizer are set to their

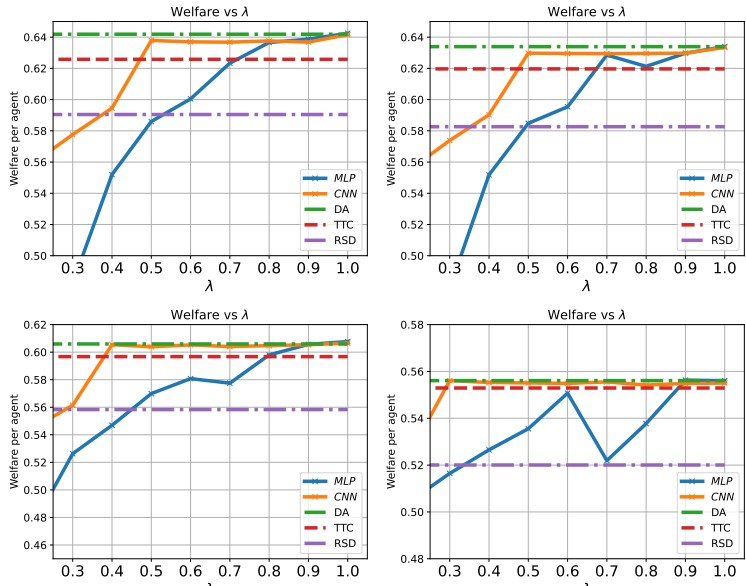

**Figure 6:** Comparing welfare per agent of the learned mechanisms (through CNNs and MLP architecture) for different values of the tradeoff parameter $\lambda$ with the best of the firm- and worker- proposing DA, as well as TTC, and RSD. The results are shown for uncorrelated preferences as well as an increasing correlation between preferences ($p_{\text{corr}} \in \{0.25, 0.5, 0.75\}$)

default values. We sample a fresh minibatch of 1024 profiles and train our neural networks for a total of 50000 minibatch iterations. We reduce the learning rate by half once at $10000^{th}$ iteration and once at $25000^{th}$ iteration.

For the CNN architecture, we use $R = 4$ hidden layers with $J = 64$ filters each. We use the leaky ReLU activation function at each of these layers. Additionally, we also make use of residual connections and instance norms. We train these network with Adam Optimizer with a learning rate of 0.001. Like the smaller setting, we sample a fresh minibatch of 1024 profiles and train our neural networks for a total of 20000 minibatch iterations.

For our training, we use a single Tesla A100 or H100 GPU per setting. For the smaller setting, it 5 hours to train. For the larger settings, it took 11 - 15 hours.

## F WELFARE

Figure 6 shows the expected welfare for the learned mechanisms, measured here for the equi-spaced utility function (the function used in the input representation) for Settings A and B with $n = m = 4$. We define the welfare of a mechanism $g$ (for the equi-spaced utility function) on a profile $\succ$ as:

$$welfare(g, \succ) = \frac{1}{2}\left(\frac{1}{n} + \frac{1}{m}\right) \sum_{w \in W} \sum_{f \in F} g_{wf}(\succ)\left(p^{\succ}_{wf} + q^{\succ}_{wf}\right). \tag{12}$$

We compare against the maximum of the expected welfare achieved by the worker- and firm-proposing DA and TTC mechanisms, as well as that from RSD. As we increase $\lambda$, and the learned mechanisms come closer to DA, the welfare of the learned mechanisms improves. It is notable that for choices of $\lambda$ in the range 0.8 and higher, i.e., the choices of $\lambda$ that provide interesting opportunities for improving SP relative to DA, we also see good welfare. We also see that TTC, and especially RSD have comparably lower welfare. It bears emphasis that when a mechanism is not fully SP, as is the case for all mechanisms except RSD, this is an idealized view of welfare since it assumes truthful reports. In fact, we should expect welfare to be reduced through strategic misreports and the substantially improved SP properties of the learned mechanisms relative to DA (Figure 3) would be expected to further work in favor of improving the welfare in the learned mechanisms relative

to DA.[3] Lastly, we observe that for small values of $\lambda$ the learned mechanisms have relatively low welfare compared to RSD. This is interesting and suggests that achieving IR together with SP (recall that RSD is not IR!) is very challenging in two-sided markets.

---

[3]In fact, the same is true for the stability of a non-SP mechanism such as DA, but it has become standard to assume truthful reports to DA in considering the stability properties of DA.

