# OpenReview forum: "Deep Learning for Two-Sided Matching"
_ICLR.cc/2025/Conference — Submitted to ICLR 2025_

### Official Review · Reviewer_23gR · 2024-10-17

**Soundness:** 4
**Presentation:** 3
**Contribution:** 3
**Rating:** 6
**Confidence:** 4

**Summary:**

This paper presents an innovative study on using deep learning for the automated design of two-sided matching mechanisms. The core contribution is the exploration of the tradeoff between strategy-proofness and stability, two critical properties that cannot be fully achieved simultaneously in traditional matching mechanisms. The authors introduce novel differentiable surrogates to quantify ordinal strategy-proofness and stability, training neural networks to generate randomized matchings. The results show that the efficient frontier, characterized by the learned mechanisms, outperforms traditional baselines like Deferred Acceptance (DA), Top Trading Cycles (TTC), and Random Serial Dictatorship (RSD). This approach sets a new direction for economic theory, especially in the design of mechanisms that improve on current tradeoffs.

**Strengths:**

1 The use of deep learning to design two-sided matching mechanisms is highly innovative and represents a significant shift from traditional economic approaches. By employing neural networks, the paper opens up new possibilities for discovering previously unknown tradeoffs between stability and strategy-proofness.

2 The paper successfully demonstrates that deep learning can uncover a superior tradeoff frontier compared to traditional convex combinations of existing mechanisms (DA, TTC, and RSD). This is a substantial contribution, highlighting the potential of machine learning in economic theory and mechanism design.

3 The introduction of both fully connected and convolutional neural network architectures, especially the CNN designed for matching mechanisms using 1×1 convolutions, shows a thoughtful adaptation of neural networks for this domain. The use of permutation equivariance in the CNN design is particularly commendable, as it enhances the generalization and scalability of the approach.

**Weaknesses:**

1 One major limitation of the proposed method is its scalability. While the paper claims to scale efficiently to markets with up to 50 agents on each side, this size is relatively small for many real-world applications. In large-scale matching markets, such as job markets or school admissions, the number of participants can easily reach into the thousands. Therefore, the current methodology may struggle to handle the complexity of large-scale matching problems, where the number of workers and firms is significantly higher. The approach should be optimized to better handle larger scales, with at least the capability to manage thousands of participants on both sides of the market.

2 The lack of publicly available datasets for testing the approach is a notable shortcoming. While the authors attempt to capture real-world structures through simulations, real-world validation remains essential to prove the practical viability of the proposed mechanisms. Without such validation, the generalizability of the findings remains uncertain.

3 The adversarial learning approach, where defeating misreports are generated to quantify violations of first-order stochastic dominance (FOSD), may introduce significant computational overhead. This could become more problematic as the size of the market increases, further exacerbating scalability issues.

**Questions:**

1 How do the authors envision scaling this framework to larger markets with thousands of agents on each side? Are there specific techniques or optimizations being considered to extend the approach beyond the current limit of 50 agents? one good way should be as follows. Considering the scalability challenges, is it possible to reduce the computational complexity by focusing on truncated preference lists rather than allowing for empty sets? In real-world markets with thousands of workers and firms, where each participant's preference list is limited to a fixed number (e.g., 10 items), using truncated preferences could significantly reduce the matching time. Could this modification lead to more efficient random matching generation while maintaining strong tradeoffs between stability and strategy-proofness?

2 Given the limited availability of public datasets, what kinds of real-world matching problems do the authors envision their approach being most applicable to? Are there any ongoing efforts to test this framework in partnership with real-world markets?Have you ever considered partnering with educational institutions or job placement agencies to obtain anonymized matching data, or inquired about potential synthetic datasets that might closely mimic real-world scenarios?


3  Is there a better way to design the loss function that could more effectively balance the tradeoff between strategy-proofness and stability, or perhaps other properties? Could alternative loss formulations lead to better performance or more efficient training?

---

> ### Author Response · Authors · 2024-11-22
>
> We thank you for the thoughtful review and constructive feedback.
>
> **Scalability:**
> We acknowledge that scalability to larger markets is an important challenge. While our current implementation focuses on markets with up to 50 agents per side by restricting the support of the preference distributions, we appreciate your suggestion to use truncated preference lists. This approach could effectively reduce the size of preference domain for misreports and allow for additional scaling.
>
> At the same time, the market size addressed in this study is significantly larger than that explored in prior work. Moreover, insights and mechanisms developed for smaller markets can be applied to larger settings by decomposing them into smaller zones or segments. For instance, in a workers-firms scenario, a country-wide job market could be segmented into smaller metro areas, with each segment solved independently. This locality-based approach aligns with practical constraints.
>
> ---
>
> **Real-World Problems:**
> Please refer to the comment [here](https://openreview.net/forum?id=p1HeFnn2AA&noteId=URVrpIqEWh)
>
> ---
>
> **Loss functions:**
> Our approach employs the strongest notions of stability (strong-stability) and strategy-proofness (SP), specifically using first-order stochastic dominance (FOSD).  Relaxing these notions to study trade-offs with weaker variants could be a valuable direction for future exploration.
>
> To balance the multi-objective trade-off between stability and SP, we utilize a linear scalarization method with the trade-off parameter $\lambda$. This method is effective and computationally scalable. Investigating alternative multi-objective optimization techniques or adaptive methods to precisely target specific regions of the frontier represents an exciting avenue for future research.

---

> > ### Comment · Reviewer_23gR · 2024-11-29
> > **scalability**
> >
> > Thanks for your response. For the scalability method you mentioned is interesting,  however I don't think it could cover many cases, as the original college enrollment problem. I will keep the score.

---

### Official Review · Reviewer_ohtn · 2024-10-25

**Soundness:** 3
**Presentation:** 3
**Contribution:** 3
**Rating:** 8
**Confidence:** 3

**Summary:**

The paper proposes a deep learning pipeline to study the tradeoff between strategy-proofness and stability in two-sided matching markets. The paper formulates SP violation and stability violation quantitatively as a machine learning problem. Extensive experiments are conducted to verify the claim.

**Strengths:**

The idea of studying the tradeoff between SP and stability with deep learning is novel and intuitive. The paper opens up a new direction in mechanism design with machine learning techniques. I generally believe that this is a good paper with novel ideas and clear presentation.

**Weaknesses:**

The experiment is conducted on synthetic data as compared to real-world data.

**Questions:**

The problem has the assumption that the market is one-to-one, and there are no ties. How will the problem formulation be different if these assumptions are removed?

---

> ### Author Response · Authors · 2024-11-22
>
> Thank you for your thoughtful review and recognition of our work's novelty and contribution to mechanism design.
>
> For concerns regarding real-world data, please refer to the comment [here](https://openreview.net/forum?id=p1HeFnn2AA&noteId=URVrpIqEWh).
>
> ---
>
> >The problem has the assumption that the market is one-to-one, and there are no ties. How will the problem formulation be different if these assumptions are removed?
>
> **Ties:**
> We do allow for ties. Randomization (and the CNN architecture) ensures that agents that have similar preferences and are similarly liked by other firms will have equal marginal probabilities over their outcomes.
>
> **Extensions to many-to-one Matching:**
> In many-to-one matching problems, the row sums of the output must be at most 1 (workers matched to one firm), and the column sums must not exceed the capacities of the firms. To ensure valid matchings, we output two sets of scores: one with row sums normalized to 1 and another with column sums normalized to 1, scaled by firm capacities. The final matching is the element-wise minimum of these scores, satisfying all constraints. (Section 4.1, under outputs)
>
> The Birkhoff-von Neumann theorem applies only to one-to-one matchings. However, Theorem 1 in Budish et al. (2013) guarantees that an output structure that  adheres to the bi-hierarchy structure proposed in that paper corresponds to a distribution over deterministic many-to-one assignments with marginal probabilities equal to the neural network output. This can be used to extend our work to many-to-one matchings.
>
> The loss functions we use are compatible with many-to-one settings but require adjustments for non-wastefulness in stability violations, where residuals should be computed as  $q_f - \sum_w g_{wf}$ where $q_f$ is the capacity for firm ‘f’. For strategyproofness (SP) violations, we can model firms potentially misreporting their capacities by treating (in addition to misreporting the preferences) by taking into account the capacity as an input to the neural network.  This input would be used exclusively for normalization in the output layer.

---

### Official Review · Reviewer_guf5 · 2024-11-03

**Soundness:** 3
**Presentation:** 3
**Contribution:** 3
**Rating:** 8
**Confidence:** 4

**Summary:**

This paper proposes a deep learning-based mechanism for two-sided matching that explores the Pareto frontier between strategy-proofness (SP) and stability—two properties that are theoretically impossible to satisfy simultaneously. The authors parameterize the matching mechanism using a neural network that takes participants' reported preferences from both sides as input and outputs a randomized matching result. The loss function is designed as a convex combination of SP and stability violations. Experimental results demonstrate that the proposed method finds a more efficient frontier than classical mechanisms.

**Strengths:**

1. The paper initiates the study of using neural networks to discover two-sided mechanisms from data. The idea of employing neural networks for mechanism learning is compelling.
2. The learned mechanism is capable of identifying a more efficient frontier than traditional approaches.
3. The paper establishes a bridge between deep learning and traditional algorithmic game theory, which I believe is interesting to ICLR community.

**Weaknesses:**

1. The experiments are limited to two prior distributions of preferences; incorporating additional distributions would be beneficial.
2. There are no diagrams to illustrate the architecture of the proposed neural network.

**Questions:**

The market settings are unclear to me. Could you formally describe how the agents' preferences are generated? I didn't understand how to truncate the preference order, and what is the common preference order for firms (or workers).

---

> ### Author Response · Authors · 2024-11-22
>
> Thank you for your thoughtful review and positive assessment of our work.
>
> **Diagrams:**
> Thanks for suggesting this. We are currently working on one and will incorporate your feedback into our paper.
>
> ---
>
> **Explanation of preferences generation for different market settings:**
>
> Thanks for this question. We will improve our exposition to improve clarity.
> As a reminder: A dataset is a collection of $\ell$ profiles. A profile is a collection of  $m$ + $n$ rank orders.
>
> _Uncorrelated Preferences_
> Here’s how we generate a rank order for the uncorrelated preference. For each profile in the dataset, for each firm/worker:
> - First we sample a complete rank order (A complete order over 4 workers may look like this: $w_1 \succ w_2 \succ w_3 \succ w_4 \succ \bot$
> - With 20% chance (if $p_{trunc}$ = 0.2), such rank orders are to be truncated.
> - If a rank order is chosen to be truncated, a truncation length is chosen at random. For example, if the above profile is chosen to be truncated, a truncation length picked at random is say 2, the above profile becomes $w_1 \succ w_2 \succ \bot$.
>
> _Correlated Preferences_
> Here’s how we generate our preferences for the uncorrelated preference setting
> - Generate a dataset of uncorrelated preferences as above.
> - For each profile, sample a new rank order for the worker. (This new rank order is sampled uniformly at random over all possible rank orders).
> - With 75% chance (if $p_{corr}$ = 0.75), replace each of the worker’s rank order with the common order sampled in the step above.
>
> Higher the $p_{corr}$, higher the likelihood that all the workers have the same preference order within a profile.

---

> > ### Comment · Reviewer_guf5 · 2024-11-27
> >
> > Thank you for your reply that answers my question. I will keep my score.

---

> > > ### Author Response · Authors · 2024-11-27
> > >
> > > Thanks once again for the review and feedback! We have just updated the paper with the diagrams. The 1 x 1 convolution block is shown Figure 1. The overall architecture using CNNs is shown in Figure 2.

---

### Author Response · Authors · 2024-11-22
**Real-World Validation**

Public datasets for two-sided matching markets are limited, which presents a challenge with real-world validation. We thank reviewer 23gR for the suggestion to actively explore partnerships with educational institutions and job placement platforms to access anonymized data for validation purposes. This could indeed provide valuable insights into the applicability of our methods in real-world settings.

In the meantime, we have adopted synthetic datasets designed to reflect real-world-inspired structures, such as truncated and correlated preferences, to evaluate our mechanisms. Leveraging stylized probabilistic models and simulations is a well-established practice in the academic study of two-sided matching markets. We have referenced several influential studies in the paper that employ similar approaches to validate mechanism design in the absence of real-world datasets.

Moreover, our findings from these synthetic datasets offer promising directions for future real-world applications. For instance, as we increase the focus on SP in our models, we observe a corresponding increase in the need for randomization in the learned mechanisms. Such insights and conjectures could form the basis for developing new theoretical designs and practical implementations, bridging the gap between simulation and real-world applicability.

While direct real-world validation is a future goal, our current efforts using well-accepted simulation practices provide a solid foundation for advancing mechanism design. We are optimistic that these insights will translate effectively to practical settings as more opportunities for real-world testing become available.

---

### Meta-Review · Area_Chair_h23s · 2024-12-22

**Metareview:**

This is a cute paper on using deep-learning to design some data-driven matching strategy.

The reviewers rather enjoyed this paper, but I am a bit concerned by the technical contributions. This approach has been investigated in many other economic similar contexts (namely auctions, where the one of the first papers introducing MyersonNet has created a large literature).

The theoretical contributions are very thin, hence we should focus on the practical and technical ones. And on those questions, I have troubles being excited about the "novelty" as I find this paper rather incremental. It is very cute and interesting, but quite incremental, in particular, this topic has already a bunch of papers published (see eg the chapter Machine Learning for Matching Markets).

All in all, I do not think the contributions are sufficient enough with respect to the literature to reach the ICLR acceptance bar

**Additional Comments On Reviewer Discussion:**

I have read the paper myself as I find the topic particularly relevant, and I am an expert in this area.

After reading it, I have decided to go against the score of the reviewers, for the scientific reasons mentioned above, and also because I have much more expertise than them on this topic (unfortunately, this paper ended it being reviewed by three "first-time reviewers").

I know this might not be very pleasant for the authors (and I apologize for this), but this is the role of Area Chair.

---

### Decision · Program_Chairs · 2025-01-22

Reject